# Pay Attention to MLPs

**Hanxiao Liu, Zihang Dai, David R. So, Quoc V. Le**
Google Research, Brain Team
{hanxiaol,zihangd,davidso,qvl}@google.com

## Abstract

Transformers [1] have become one of the most important architectural innovations in deep learning and have enabled many breakthroughs over the past few years. Here we propose a simple network architecture, gMLP, based on MLPs with gating, and show that it can perform as well as Transformers in key language and vision applications. Our comparisons show that self-attention is not critical for Vision Transformers, as gMLP can achieve the same accuracy. For BERT, our model achieves parity with Transformers on pretraining perplexity and is better on some downstream NLP tasks. On finetuning tasks where gMLP performs worse, making the gMLP model substantially larger can close the gap with Transformers. In general, our experiments show that gMLP can scale as well as Transformers over increased data and compute.

## 1 Introduction

Transformers [1] have enabled many breakthroughs in natural language processing (e.g., [2, 3, 4, 5, 6]) and have been shown to work well for computer vision (e.g., [7, 8, 9, 10]). Thanks to this success, Transformers have largely replaced LSTM-RNN [11] as the default architecture in NLP, and have become an appealing alternative to ConvNets [12, 13, 14, 15, 16, 17] in computer vision.

The Transformer architecture combines two important concepts: (1) a recurrent-free architecture which computes the representations for each individual token in parallel, and (2) multi-head self-attention blocks which aggregate spatial information across tokens. On one hand, the attention mechanism [18] introduces the inductive bias that the spatial interactions should be dynamically parameterized based on the input representations. On the other hand, it is known that MLPs with static parameterization can represent arbitrary functions [19]. It therefore remains an open question whether the inductive bias in self-attention is essential to the remarkable effectiveness of Transformers.

Here we study the necessity of self-attention modules in key language and vision applications of Transformers. Specifically, we propose an MLP-based alternative to Transformers without self-attention, which simply consists of channel projections and spatial projections with static parameterization. We experiment with several design choices for this architecture and find spatial projections work well when they are linear and paired with multiplicative gating (Figure 1). We name the model **gMLP** because it is built out of basic MLP layers with gating.

We apply gMLP to image classification and obtain strong results on ImageNet. gMLP achieves comparable performance with DeiT [8], namely Vision Transformer (ViT) [7] with improved regularization, in a similar training setup. With 66% less parameters, a gMLP model is 3% more accurate than MLP-Mixer [20]. Together with Tolstikhin et al. [20], Melas-Kyriazi [21], Touvron et al. [22] and Ding et. al. [23], our results question the necessity of self-attention layers in Vision Transformers.

We apply gMLP to masked language modeling (MLM) in the BERT [2] setup, one of the most well-established applications of Transformers, and find that it is as good as Transformers at minimizing perplexity during pretraining. Our experiments indicate that perplexity is only correlated with model capacity and is insensitive to the presence of self-attention. As capacity increases, we observe that

Pseudo-code for the gMLP block

```
def gmlp_block(x, d_model, d_ffn):
  shortcut = x
  x = norm(x, axis="channel")
  x = proj(x, d_ffn, axis="channel")
  x = gelu(x)
  x = spatial_gating_unit(x)
  x = proj(x, d_model, axis="channel")
  return x + shortcut

def spatial_gating_unit(x):
  u, v = split(x, axis="channel")
  v = norm(v, axis="channel")
  n = get_dim(v, axis="spatial")
  v = proj(v, n, axis="spatial", init_bias=1)
  return u * v
```

Figure 1: Overview of the gMLP architecture with Spatial Gating Unit (SGU). The model consists of a stack of $L$ blocks with identical structure and size. All projection operations are linear and "$\odot$" refers to element-wise multiplication (linear gating). The input and output protocols follow BERT for NLP and ViT for vision. Unlike Transformers, gMLPs do not require positional encodings, nor is it necessary to mask out the paddings during NLP finetuning.

both pretraining and finetuning metrics for gMLPs improve as quickly as for Transformers. This is remarkable because it indicates gMLPs scale just as well as Transformers despite the absence of self-attention, and any performance gap can always be offset by training a larger model with increased data and compute. With a standard 256-batch size $\times$ 1M-step training setup as in original BERT, a large gMLP model achieves 87.7% accuracy on MNLI and 82.1% F1 on SQuAD v2.0. Note, these are better than the BERT$_{\text{large}}$ results reported in Devlin et al. [2] obtained using Transformers.

For BERT's finetuning, Transformers can be more practically advantageous over gMLPs on tasks that require cross-sentence alignment (e.g., by 0.8% on MNLI-m in the 300M-param regime), even with similar pretraining perplexity. This problem can be addressed by making gMLPs substantially larger—3$\times$ as large as Transformers. A more practical solution is to blend in only a tiny bit of self-attention—a single-head self-attention with size up to 128 is sufficient to make gMLPs outperform Transformers on all NLP tasks we evaluated with even better parameter efficiency. The improvement is sometimes very significant (e.g., +4.4% on SQuAD v2.0 over BERT$_{\text{large}}$).

Overall, the surprising effectiveness of gMLPs in both vision and NLP domains suggests that self-attention is not a necessary ingredient for scaling up machine learning models, although it can be a useful addition depending on the task. With increased data and compute, models with simpler spatial interaction mechanisms such as gMLP can be as powerful as Transformers and the capacity allocated to self-attention can be either removed or substantially reduced.

## 2  Model

Our model, gMLP, consists of a stack of $L$ blocks with identical size and structure. Let $X \in \mathbb{R}^{n \times d}$ be the token representations with sequence length $n$ and dimension $d$. Each block is defined as:

$$Z = \sigma(XU), \qquad \tilde{Z} = s(Z), \qquad Y = \tilde{Z}V \tag{1}$$

where $\sigma$ is an activation function such as GeLU [24]. $U$ and $V$ define linear projections along the channel dimension—the same as those in the FFNs of Transformers (e.g., their shapes are $768 \times 3072$ and $3072 \times 768$ for BERT$_{\text{base}}$). Shortcuts, normalizations and biases are omitted for brevity.

A key ingredient in the aforementioned formulation is $s(\cdot)$, a layer which captures spatial interactions (see below). When $s$ is an identity mapping, the above transformation degenerates to a regular FFN, where individual tokens are processed independently without any cross-token communication. One of our major focuses is therefore to design a good $s$ capable of capturing complex spatial interactions across tokens. The overall block layout is inspired by inverted bottlenecks [25] which define $s(\cdot)$ as a spatial depthwise convolution. Note, unlike Transformers, our model *does not require position embeddings* because such information will be captured in $s(\cdot)$.

Our model uses exactly the same input and output protocols as BERT (for NLP) and ViT (for vision). For example, when finetuning on language tasks, we concatenate together multiple text segments followed by paddings, and the predictions are deduced from the last-layer representation of a reserved `<cls>` symbol. Although many of these protocols were introduced for Transformers and hence can be suboptimal for gMLPs, strictly following them helps avoid confounding factors in our experiments and makes our layers more compatible with existing Transformer implementations.

## 2.1 Spatial Gating Unit

To enable cross-token interactions, it is necessary for the layer $s(\cdot)$ to contain a contraction operation over the spatial dimension. The simplistic option would be a linear projection:

$$f_{W,b}(Z) = WZ + b \tag{2}$$

where $W \in \mathbb{R}^{n \times n}$ is a matrix for which the size is the same as the sequence length, $n$, and $b$ refers token-specific biases. For example, if the padded input sequence has 128 tokens, the shape for $W$ will be 128×128. Unlike self-attention where $W(Z)$ is dynamically generated from $Z$, the spatial projection matrix $W$ here in Equation (2) is independent from the input representations.

In this work, we formulate layer $s(\cdot)$ as the output of linear gating:

$$s(Z) = Z \odot f_{W,b}(Z) \tag{3}$$

where $\odot$ denotes element-wise multiplication. For training stability, we find it critical to initialize $W$ as near-zero values and $b$ as ones, meaning that $f_{W,b}(Z) \approx \mathbf{1}$ and therefore $s(Z) \approx Z$ at the beginning of training. This initialization ensures each gMLP block behaves like a regular FFN at the early stage of training, where each token is processed independently, and only gradually injects spatial information across tokens during the course of learning.

We further find it effective to split $Z$ into two independent parts $(Z_1, Z_2)$ along the channel dimension for the gating function and for the multiplicative bypass:

$$s(Z) = Z_1 \odot f_{W,b}(Z_2) \tag{4}$$

We also normalize the input to $f_{W,b}$ which empirically improves stability of large NLP models. This gives us the unit illustrated in Figure 1, which we refer to as the *Spatial Gating Unit* (SGU) in the rest of the paper. In Table 3, we provide ablation studies to compare SGU with several other variants of $s(\cdot)$, showing that it works better and narrows the performance gap with self-attention.

**Connections to Existing Layers.** The overall formulation of SGU resembles Gated Linear Units (GLUs) [26, 27, 28] as well as earlier works including Highway Networks [29] and LSTM-RNNs [11]. A key distinction is that our gating is computed based on a projection over the spatial (cross-token) dimension rather than the channel (hidden) dimension. SGU is also related to Squeeze-and-Excite (SE) blocks [30] in terms of element-wise multiplication. However, different from SE blocks, SGU does not contain cross-channel projections at all, nor does it enforce permutation invariance (a key feature for content-based attentive modules) due to its static parameterization for the spatial transformation. The spatial projection in SGU could in theory learn to express superficial depthwise convolutions—unlike typical depthwise convolutions with channel-specific filters, SGU learns only a single transformation shared across channels. Finally, we note SGUs offer an alternative mechanism to capture high-order relationships other than self-attention. Specifically, the output for Equation (3) contains up to 2nd-order interactions (e.g., $z_i z_j$) whereas output for self-attention (assuming no nonlinearity) contains up to 3rd-order interactions (e.g., $q_i k_j v_k$). In terms of computation cost, SGU has $n^2 e/2$ multiply-adds which is comparable to the $2n^2 d$ of dot-product self-attention.[1] Both are linear over the input channel size and quadratic over the sequence length $n$.

## 3 Image Classification

Here we examine gMLP in the vision domain by applying it to the image classification task on ImageNet [31] without using extra data. We compare our MLP-like models with recent attentive

---

[1]The input channel size $e$ for SGU is typically larger than the input channel size $d$ for self-attention, because the former is applied in the middle of the block after a channel expansion.

models based on vanilla Transformers, including Vision Transformer (ViT) [7], DeiT [8] (ViT with improved regularization), and several other representative convolutional networks.

Table 1 summarizes the configurations of our gMLP image classification models. The input and output protocols follow ViT/B16 where the raw image is converted into 16×16 patches at the stem. The depth and width are chosen so that the models are comparable with ViT/DeiT in capacity. Like Transformers, we find gMLPs tend to drastically overfit the training data. We therefore apply a similar regularization recipe as the one used in DeiT.[2] To avoid extensive tuning, we adjust only the strengths of stochastic depth [32] as we move from smaller to larger models in Table 1. All the other hyperparameters remain shared across our three models. See Appendix A.1 for details.

Table 1: Architecture specifications of gMLP models for vision.

| | #L | $d_{\mathrm{model}}$ | $d_{\mathrm{ffn}}$ | Params (M) | FLOPs (B) | Survival Prob |
|---|---|---|---|---|---|---|
| gMLP-Ti | 30 | 128 | 768 | 5.9 | 2.7 | 1.00 |
| gMLP-S | 30 | 256 | 1536 | 19.5 | 8.9 | 0.95 |
| gMLP-B | 30 | 512 | 3072 | 73.4 | 31.6 | 0.80 |

Our ImageNet results are summarized in Table 1 and Figure 2.[3] It is interesting to see that gMLPs are comparable with DeiT [8], namely ViT [7] trained using improved regularization. The results suggest that models without self-attention can be as data-efficient as Transformers for image classification. In fact, when the models are properly regularized, their accuracies seem better correlated with capacity instead of the presence of self-attention. Moreover, the accuracy-parameter/FLOPs tradeoff of gMLPs surpasses all concurrently proposed MLP-like architectures [20, 21, 22], which we attribute to the effectiveness of our Spatial Gating Unit (see Table 3 in the next section for an ablation). We also note while gMLPs are competitive with vanilla Transformers, their performance is behind the best existing ConvNet models (e.g., [33, 34]) or hybrid models (e.g., [35, 36, 37, 38, 10]).

Table 2: ImageNet-1K results without extra data.

| Model | ImageNet Top-1 (%)* | Input Resolution | Params (M) | MAdds (B) |
|---|---|---|---|---|
| | ConvNets | | | |
| ResNet-152 [16] | 78.3 | 224 | 60 | 11.3 |
| RegNetY-8GF [39] | 81.7 | 224 | 39 | 8.0 |
| EfficientNet-B0 [17] | 77.1 | 224 | 5 | 0.39 |
| EfficientNet-B3 [17] | 81.6 | 300 | 12 | 1.8 |
| EfficientNet-B7 [17] | 84.3 | 600 | 66 | 37.0 |
| NFNet-F0 [33] | 83.6 | 192 | 72 | 12.4 |
| | Transformers | | | |
| ViT-B/16 [7] | 77.9 | 384 | 86 | 55.4 |
| ViT-L/16 [7] | 76.5 | 384 | 307 | 190.7 |
| DeiT-Ti [8] (ViT+reg) | 72.2 | 224 | 5 | 1.3 |
| DeiT-S [8] (ViT+reg) | 79.8 | 224 | 22 | 4.6 |
| DeiT-B [8] (ViT+reg) | 81.8 | 224 | 86 | 17.5 |
| | MLP-like† | | | |
| Mixer-B/16 [20] | 76.4 | 224 | 59 | 12.7 |
| Mixer-B/16 (our setup) | 77.3 | 224 | 59 | 12.7 |
| Mixer-L/16 [20] | 71.8 | 224 | 207 | 44.8 |
| ResMLP-12 [22] | 76.6 | 224 | 15 | 3.0 |
| ResMLP-24 [22] | 79.4 | 224 | 30 | 6.0 |
| ResMLP-36 [22] | 79.7 | 224 | 45 | 8.9 |
| gMLP-Ti (ours) | 72.3 | 224 | 6 | 1.4 |
| gMLP-S (ours) | 79.6 | 224 | 20 | 4.5 |
| gMLP-B (ours) | 81.6 | 224 | 73 | 15.8 |

\* Standard deviation across multiple independent runs is around 0.1.
† Tokenization & embedding process at the stem can be viewed as a convolution.

Figure 3 visualizes the spatial projection matrices in gMLP-B. Remarkably, the spatial weights after learning exhibit both locality and spatial invariance. In other words, each spatial projection matrix effectively learns to perform convolution with a data-driven, irregular (non-square) kernel shape.

---

[2]Unlike DeiT, we do not use repeated augmentation or random erasing.

[3]Additional results on ImageNet-21K and COCO are available in Appendix E.

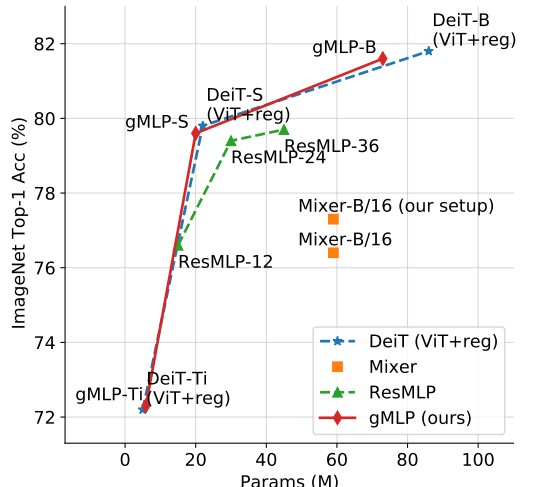

Figure 2: ImageNet accuracy vs model capacity.

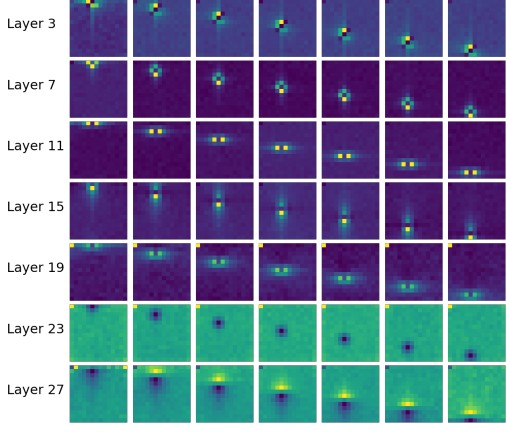

Figure 3: Spatial projection weights in gMLP-B. Each row shows the filters (reshaped into 2D) for a selected set of tokens in the same layer.

# 4 Masked Language Modeling with BERT

Here we conduct empirical studies over the masked language modeling (MLM) task. The input/output protocol for both pretraining and finetuning follows BERT [2]. Different from Transformer-based models, we do not use positional encodings. We also find it unnecessary to mask out `<pad>` tokens in gMLP blocks during finetuning as the model can quickly learn to ignore them. For ablations and case studies, all models are trained with batch size 2048, max length 128 for 125K steps over the RealNews-like subset of C4 [5]. For main results, models are trained with batch size 256, max length 512 for 1M steps over the full English C4 dataset. See Appendix A.2 for details.

Our preliminary MLM experiments show that gMLPs always learn Toeplitz-like matrices as the spatial weights (Appendix C). This means gMLPs are able to learn the notion of shift invariance from data, a property naturally implied by the MLM task where any offset of the input sequence does not affect the slot filling outcome. In this case, the learned $f_{W,b}(\cdot)$ acts like a 1-d convolution whose kernel size equals the entire sequence length (unlike depthwise convolution with channel-specific filters, here the same $W$ is shared across channels). In the following MLM experiments, we restrict $W$ to be a Toeplitz matrix to avoid redundant model parameterization (since $W$ will be Toeplitz-like regardless after learning). Note this constraint is empirically quality-neutral.

## 4.1 Ablation: The Importance of Gating in gMLP for BERT's Pretraining

In Table 3 below, we establish baselines for our ablation studies. These include:

1. BERT with a Transformer architecture and learnable absolute position embeddings.

2. BERT with a Transformer architecture and T5-style learnable relative position biases [5]. The biases are both layer- and head-specific as we find this yields the best results.

3. Same as above, but we remove all content-dependent terms inside the softmax and only retain the relative positional biases. This baseline is a straightforward variant of Transformers without self-attention, which can also be viewed as a Random Synthesizer [40].

4. MLP-Mixer [20] which replaces the multi-head self-attention module in Transformers with a two-layer spatial MLP. This model was developed for image classification and here we investigate it on MLM tasks using the same training setup with BERT and gMLP.

We compare these baselines against several versions of gMLPs with similar sizes in Table 3. Note that Multiplicative, Split (last row) is the Spatial Gating Unit we describe in the method section and use in the rest of the paper. First, SGU outperforms other variants in perplexity. Secondly and remarkably, gMLP with SGU also achieves perplexity comparable to Transformer. Note the difference between the strongest baseline (perplexity=4.26) and ours (perplexity=4.35) is insignificant relative to the

Table 3: MLM validation perplexities of Transformer baselines and four versions of gMLPs. $f$ refers to the spatial linear projection in Equation (2) with input normalization. The MLP-Mixer baseline model has L=24 layers with $d_{\text{model}}$=768, $d_{\text{spatial}}$=384 and $d_{\text{ffn}}$=3072. Each gMLP model has L=36 layers with $d_{\text{model}}$=512 and $d_{\text{ffn}}$ = 3072. No positional encodings are used for Mixer or gMLPs.

| Model | Perplexity* | Params (M) |
|---|---|---|
| BERT$_{\text{base}}$ | 4.37 | 110 |
| BERT$_{\text{base}}$ + rel pos | 4.26 | 110 |
| BERT$_{\text{base}}$ + rel pos - attn | 5.64 | 96 |
| MLP-Mixer | 5.34 | 112 |
| Linear gMLP, $s(Z) = f(Z)$ | 5.14 | 92 |
| Additive gMLP, $s(Z) = Z + f(Z)$ | 4.97 | 92 |
| Multiplicative gMLP, $s(Z) = Z \odot f(Z)$ | 4.53 | 92 |
| Multiplicative, Split gMLP, $s(Z) = Z_1 \odot f(Z_2), Z = Z_1 \| Z_2$ | 4.35 | 102 |

\* Standard deviation across multiple independent runs is around 0.01.

perplexity change when the models are scaled (see Table 4 in the next section). Spatial projection weights learned by gMLPs are visualized in Figure 4.

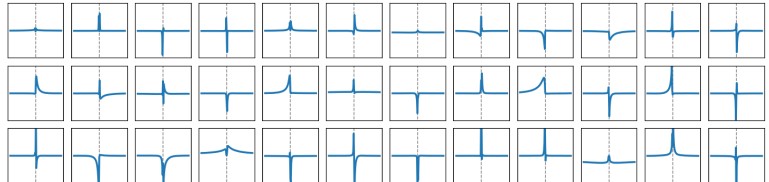

Figure 4: Visualization of the spatial filters in gMLP learned on the MLM task. For each layer in the model we plot the row in $W$ associated with the token in the middle of the sequence. The x-axis of each subplot has a length of 128 which equal the number of tokens in the sequence. The learned filters appear to be smooth and have several types: forward-looking (e.g., 1st in 2nd row), backward-looking (e.g., 5th in 2nd row) and bi-directional (e.g., 2nd last in the last row).

## 4.2   Case Study: The Behavior of gMLP as Model Size Increases

In Table 4, we investigate the scaling properties of Transformers and gMLPs in BERT as their model capacity grows. Specifically, we scale the depth of these models by a factor of $\{0.5, 1, 2, 4\}\times$ and report the their pretraining MLM perplexities on the validation set as well as finetuning results on the dev sets of two tasks in GLUE [41]. Note each individual Transformer layer is effectively two consecutive blocks: one for self-attention and one for FFN. In the table below we use the notation of 12 + 12 to refer to 12 of self-attention blocks plus 12 of FFN blocks in the Transformer baselines.

Table 4: Pretraining and dev-set finetuning results over increased model capacity. We use the relative positional encoding scheme for Transformers which performs the best in Table 3.

| Model | #L | Params (M) | Perplexity | SST-2 | MNLI-m |
|---|---|---|---|---|---|
| Transformer | 6+6 | 67 | **4.91** | 90.4 | 81.5 |
| gMLP | 18 | 59 | 5.25 | 91.2 | 77.7 |
| Transformer | 12+12 | 110 | **4.26** | 91.3 | 83.3 |
| gMLP | 36 | 102 | 4.35 | 92.3 | 80.9 |
| Transformer | 24+24 | 195 | 3.83 | 92.1 | 85.2 |
| gMLP | 72 | 187 | **3.79** | 93.5 | 82.8 |
| Transformer | 48+48 | 365 | 3.47 | 92.8 | 86.3 |
| gMLP | 144 | 357 | **3.43** | 95.1 | 84.6 |

The results above show that a deep enough gMLP is able to match and even outperform the perplexity of Transformers with comparable capacity.[4] In addition, the perplexity-parameter relationships for

---

[4]We also experimented with deeper-and-thinner Transformers (with capacity fixed) but found increasing depth further does not improve perplexity. See Appendix B for more details.

both architecture families approximately follow a power law (left of Figure 5). This implies the empirical scaling laws originally observed for Transformer-based language models [42] might be broadly applicable across different model families.

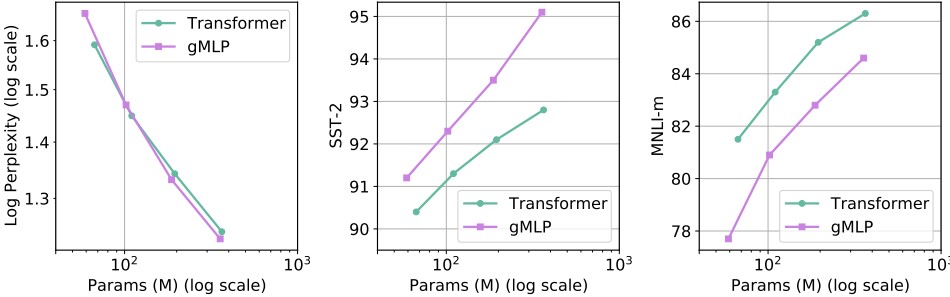

Figure 5: Scaling properties with respect to perplexity and finetuning accuracies. The figures show that for pretraining, gMLPs are equally good at optimizing perplexity as Transformers. For finetuning, the two model families exhibit comparable scalability despite task-specific offsets.

Table 4 also leads to an interesting observation that the pretraining perplexities across different model families are not equal in terms of finetuning. While gMLPs outperform Transformers on SST-2, they are worse on MNLI. The results imply that the finetuning performance for NLP tasks is a function of not only the perplexity but also the inductive bias in the architecture. Figure 5 shows that despite the architecture-specific discrepancies between pretraining and finetuning, gMLPs and Transformers exhibit comparable scalability (slope) on both finetuning tasks. This means one can always offset the gap by enlarging the model capacity. In other words, the results indicate that model scalability with respect to downstream metrics can be independent from the presence of self-attention.

### 4.3 Ablation: The Usefulness of Tiny Attention in BERT's Finetuning

So far we have found that self-attention is not a required component to achieve strong MLM perplexity or scalability. At the meantime, we also identified NLP finetuning tasks where gMLPs transfer less well than Transformers (Table 4). The fact that our MLP-like model is advantageous on SST-2 but worse on MNLI is particularly informative—the former is a single-sentence task whereas the latter involves sentence pairs (premise and hypothesis) [43]. We suspect the role of self-attention during finetuning is related to cross-sentence alignment.

To isolate the effect of self-attention, we experiment with a hybrid model where a tiny self-attention block is attached to the gating function of gMLP (Figure 6). Since gMLP itself is already capable in capturing spatial relationships, we hypothesize that this extra self-attention module does not have to be heavy, and that its presence is more relevant than its capacity. A typical tiny attention module in our experiments has only a single head with size 64, significantly smaller than a typical multi-head self-attention in Transformers with 12 heads and a total size of 768. In the following, we refer to the hybrid model, namely gMLP with a tiny self-attention, as *aMLP* ("a" for attention).

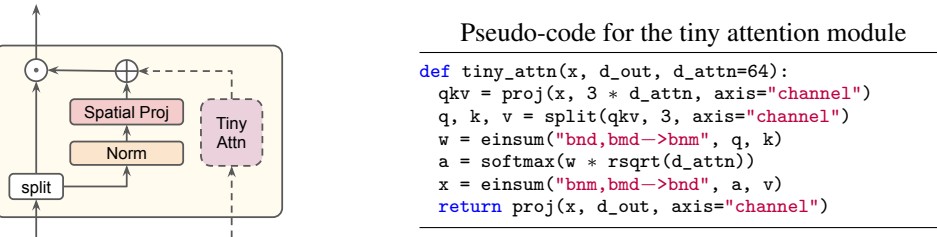

Figure 6: Hybrid spatial gating unit with a tiny self-attention module. We use the normalized input of the gMLP block (endpoint after the input normalization and right before the channel expansion) as the input to the tiny self-attention. For SGU we have $d_{\text{out}} = d_{\text{ffn}}/2$ due to the channel split.

In Figure 7, we investigate the transferability of MLM models via the calibration plots between their pretraining perplexities and finetuning metrics. Models evaluated include $\text{BERT}_{\text{base}}$, gMLP

and its hybrid version aMLP with a 64-d single-head self-attention (Figure 6). The data points were collected by varying the model depth by $\{0.5, 1, 2\}\times$ or data by $\{1, 2, 4, 8\}\times$. It can be seen that gMLPs transfer better to SST-2 than Transformers regardless of the presence of self-attention, While gMLP performs worse on MNLI, attaching a tiny bit of self-attention is sufficient to close the gap. In Appendix D we visualize the tiny self-attention modules in aMLP over MNLI examples, showing that they are primarily responsible for the alignment between sentence pairs.

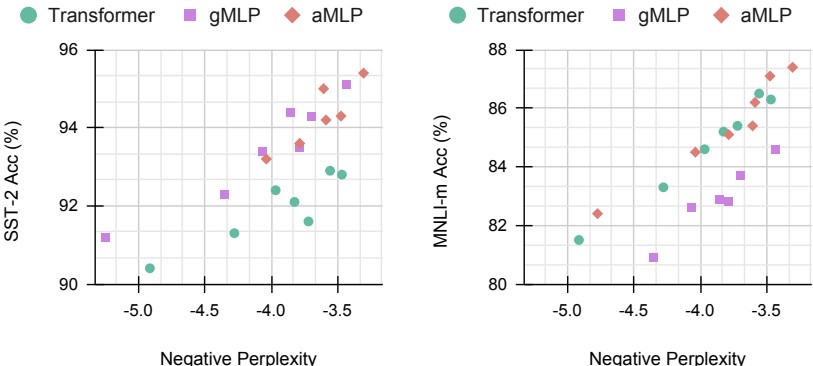

Figure 7: Transferability from MLM pretraining perpexity to finetuning accuracies on GLUE. aMLP refers to gMLP enhanced with a 64-d single-head self-attention, as illustrated in Figure 6. In contrast, each self-attention module in the BERT baseline contains 12 heads with a total size of 768.

In Figure 8 we put together the scaling properties of the three models, showing that aMLP (gMLP + tiny attention) consistently outperforms Transformer on both finetuning tasks.

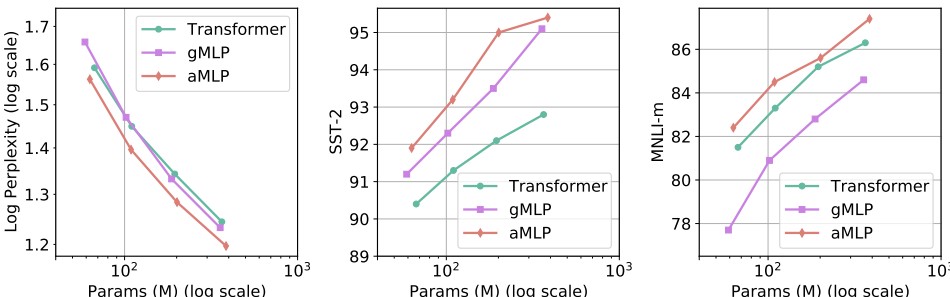

Figure 8: Comparing the scaling properties of Transformers, gMLPs and aMLPs (with 64-d, single-head attention). Results were obtained using the same setup in Section 4.2.

## 4.4 Main Results for MLM in the BERT Setup

Below we present pretraining and finetuning results in the full BERT setup. Different from ablation and case studies, here we use the full English C4 dataset and adopt a common MLM setup with batch size 256, max length 512 and 1M training steps. For fair comparison, we adjust the depth and width of gMLPs to ensure comparable model capacity with the Transformer baselines. The model specifications are given in Table 5 and hyperparameters are detailed in Appendix A.2. For finetuning, we report the dev-set performance for SST-2 and MNLI in GLUE [41] and each result entry was obtained by taking the median of five independent runs. In addition, we report finetuning results on SQuAD [44, 45] to test the models' ability in reasoning over a longer context.

Results are presented in Table 6. Consistent with our findings earlier in Section 4.1 and Section 4.2, gMLPs are competitive with Transformers in terms of perplexity, especially in the larger scale setup. There are several observations related to the finetuning results:

First, on finetuning tasks where gMLPs underperform Transformers, the performance gap tends to narrow as the model capacity increases. For example, while gMLP performs worse by 8.5% on SQuAD-v2.0 in the base scale, the performance gap relative to the baseline decreases to 2.7% at the larger scale. Notably, our gMLP$_{\text{large}}$ achieves 89.5% F1 on SQuAD-v1.1 without any self-attention or

Table 5: Model specifications in the full BERT setup.

| | Params (M) | FLOPs (B) | #L | $d_{\mathrm{model}}$ | $d_{\mathrm{ffn}}$ |
|---|---|---|---|---|---|
| BERT$_{\mathrm{base}}$ | 110 | 100.8 | 12+12 | 768 | 3072 |
| gMLP$_{\mathrm{base}}$ | 130 | 158.0 | 48 | 512 | 3072 |
| aMLP$_{\mathrm{base}}$ | 109 | 128.9 | 36 | 512 | 3072 |
| BERT$_{\mathrm{large}}$ | 336 | 341.2 | 24+24 | 1024 | 4096 |
| gMLP$_{\mathrm{large}}$ | 365 | 430.1 | 96 | 768 | 3072 |
| aMLP$_{\mathrm{large}}$ | 316 | 370.3 | 72 | 768 | 3072 |
| gMLP$_{\mathrm{xlarge}}$ | 941 | 1091.3 | 144 | 1024 | 4096 |

Table 6: Pretraining perplexities and dev-set results for finetuning. "ours" indicates models trained using our setup. We report accuracies for SST-2 and MNLI, and F1 scores for SQuAD v1.1/2.0.

| | Perplexity | SST-2 | MNLI (m/mm) | SQuAD v1.1 | v2.0 | Attn Size | Params (M) |
|---|---|---|---|---|---|---|---|
| BERT$_{\mathrm{base}}$ [2] | – | 92.7 | 84.4/- | 88.5 | 76.3 | 768 (64 × 12) | 110 |
| BERT$_{\mathrm{base}}$ (ours) | 4.17 | 93.8 | 85.6/85.7 | 90.2 | 78.6 | 768 (64 × 12) | 110 |
| gMLP$_{\mathrm{base}}$ | 4.28 | 94.2 | 83.7/84.1 | 86.7 | 70.1 | – | 130 |
| aMLP$_{\mathrm{base}}$ | 3.95 | 93.4 | 85.9/85.8 | 90.7 | 80.9 | 64 | 109 |
| BERT$_{\mathrm{large}}$ [2] | – | 93.7 | 86.6/- | 90.9 | 81.8 | 1024 (64 × 16) | 336 |
| BERT$_{\mathrm{large}}$ (ours) | 3.35 | 94.3 | 87.0/87.4 | 92.0 | 81.0 | 1024 (64 × 16) | 336 |
| gMLP$_{\mathrm{large}}$ | 3.32 | 94.8 | 86.2/86.5 | 89.5 | 78.3 | – | 365 |
| aMLP$_{\mathrm{large}}$ | 3.19 | 94.8 | 88.4/88.4 | 92.2 | 85.4 | 128 | 316 |
| gMLP$_{\mathrm{xlarge}}$ | 2.89 | 95.6 | 87.7/87.7 | 90.9 | 82.1 | – | 941 |

dynamic spatial parameterization [28], which is well above the 88.5% reported for BERT$_{\mathrm{base}}$ in Devlin et al. [2] and is only 1.4% away from the original result for BERT$_{\mathrm{large}}$. We also include one additional data point by scaling up gMLP even further. The resulting model, gMLP$_{\mathrm{xlarge}}$, outperforms BERT$_{\mathrm{large}}$ on SQuAD-v2.0—a difficult task involving question-answer pairs—without any self-attention. While this is not a fair comparison due to different model sizes, it is an existence proof that MLP-like models can be competitive with Transformers on challenging NLP tasks.

Furthermore, we show that blending in a tiny single-head self-attention of size either 64 or 128 is sufficient to make gMLPs outperform Transformers of similar capacity, sometimes by a significant margin. For example, our hybrid model aMLP$_{\mathrm{large}}$ achieves 4.4% higher F1 than Transformers on SQuAD-v2.0. The results suggest that the capacity in the multi-head self-attention of Transformers can be largely redundant, and that the majority of its functionalities can be captured by the spatial gating unit in gMLPs. The results also imply that the inductive biases in the spatial gating unit of gMLPs and the tiny attention are complementary to each other. While the benefits of architectural inductive bias may vanish over increased compute, tiny attention does improve the practical value of gMLPs in the regime that we investigate in this work.

## 5   Conclusion

Since the seminal work of Vaswani et al. [1], Transformers have been widely adopted across NLP and computer vision. This adoption has enabled many impressive results especially in NLP. To date, it is still unclear what empowers such success: is it the feedforward nature of Transformers or is it the multi-head self-attention layers in Transformers?

Our work suggests a simpler alternative to the multi-head self-attention layers in Transformers. We show that gMLPs, a simple variant of MLPs with gating, can be competitive with Transformers in terms of BERT's pretraining perplexity and ViT's accuracy. gMLPs are also comparable with Transformers in terms of the scalability over increased data and compute. As for BERT finetuning, we

find gMLPs can achieve appealing results on challenging tasks such as SQuAD without self-attention, and can significantly outperform Transformers in certain cases. We also find the inductive bias in Transformer's multi-head self-attention useful on downstream tasks that require cross-sentence alignment. However in those cases, making gMLP substantially larger closes the gap with Transformers. More practically, blending a small single-head self-attention into gMLP allows for an even better architecture without the need for increasing model size.

## Acknowledgements

We thank Gabriel Bender, Neil Houlsby, Thang Luong, Niki Parmar, Hieu Pham, Jascha Sohl-Dickstein, Noam Shazeer, Ilya Sutskever, Jakob Uszkoreit and Ashish Vaswani for their feedback.

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
