# A  Hyperparameters

## A.1  Image Classification

All ImageNet models are trained using TPUv2 with 128 cores. Each run takes 1-4 hours to complete.

|  | gMLP-Ti | gMLP-S | gMLP-B | Mixer-B |
|---|---|---|---|---|
| Stochastic depth survival prob | 1.00 | 0.95 | 0.80 | 0.95 |
| Data augmentation | | AutoAugment | | |
| Repeated Augmentation | | off | | |
| Input resolution | | 224 | | |
| Epochs | | 300 | | |
| Batch size | | 4096 | | |
| Warmup steps | | 10K | | |
| Hidden dropout | | 0 | | |
| GeLU dropout | | 0 | | |
| Attention dropout (if applicable) | | 0 | | |
| Classification dropout | | 0 | | |
| Random erasing prob | | 0 | | |
| EMA decay | | 0 | | |
| Cutmix $\alpha$ | | 1.0 | | |
| Mixup $\alpha$ | | 0.8 | | |
| Cutmix-Mixup switch prob | | 0.5 | | |
| Label smoothing | | 0.1 | | |
| Peak learning rate | | 1e-3 | | |
| Learning rate decay | | cosine | | |
| Optimizer | | AdamW | | |
| Adam $\epsilon$ | | 1e-6 | | |
| Adam $(\beta_1, \beta_2)$ | | (0.9, 0.999) | | |
| Weight decay | | 0.05 | | |
| Gradient clipping | | 1.0 | | |

Table 7: Hyperparameters for Image classification on ImageNet-1K

## A.2  Masked Language Modeling

MLM models for ablation studies are trained using TPUv3 with 32 cores. Each run takes 1-2 days to complete. Models in the full BERT setup are trained using TPUv2 with 128 cores. Each run takes 1-5 days to complete depending on the model size. The vocabulary consists of 32K cased SentencePieces.

|  | Ablation Studies | Full Results (Table 5) |
|---|:---:|:---:|
| Data | C4/RealNews | C4/English |
| Max sequence length | 128 | 512 |
| Batch size | 2048 | 256 |
| Peak learning rate | 7e-4 | 1e-4 |
| Number of steps | 125K | 1M |
| Warmup steps | 10K | |
| Hidden dropout | 0 | |
| GeLU dropout | 0 | |
| Attention dropout (if applicable) | 0 | |
| Learning rate decay | Linear | |
| Optimizer | AdamW | |
| Adam $\epsilon$ | 1e-6 | |
| Adam $(\beta_1, \beta_2)$ | (0.9, 0.999) | |
| Weight decay | 0.01 | |
| Gradient clipping | 0 | |

Table 8: Hyperparameters for MLM pretraining on C4.

|  | SST-2  MNLI | SQuAD v1.1/v2.0 |
|---|:---:|:---:|
| Max sequence length | 128 | 512 |
| Batch size | {16, 32} | 32 |
| Peak learning rate | {1e-5, 2e-5, 3e-5} | 5e-5 |
| Number of steps/epochs | 5 epochs | 8K |
| Warmup steps/portion | 10% | 1K |
| Hidden dropout | 0.1 | |
| GeLU dropout | 0 | |
| Attention dropout (if applicable) | 0.1 | |
| Learning rate decay | Linear | |
| Optimizer | AdamW | |
| Adam $\epsilon$ | 1e-6 | |
| Adam $(\beta_1, \beta_2)$ | (0.9, 0.999) | |
| Weight decay | 0.01 | |
| Gradient clipping | 0 | |

Table 9: Hyperparameters for MLM finetuning on GLUE and SQuAD.

# B  Deep-and-Thin Transformers

| Perplexity | #L | $d_{\mathrm{model}}$ | #heads | Params (M) |
|:---:|:---:|:---:|:---:|:---:|
| 4.83 | 12 + 12 | 768 | 12 | 110 |
| 5.08 | 24 + 24 | 512 | 8 | 92 |
| 4.99 | 48 + 48 | 384 | 12 | 98 |
| 5.30 | 96 + 96 | 256 | 8 | 84 |

Table 10: MLM results with increasingly deeper & thinner Transformers. As the depth increases, we adjust the model width accordingly to maintain comparable capacity. We observe that the perplexity is insensitive to the model depth at a fixed capacity, and worsens beyond 48 layers. Note these results were obtained using a similar yet different training setup from the rest of the paper.

# C  Shift Invariance in MLM

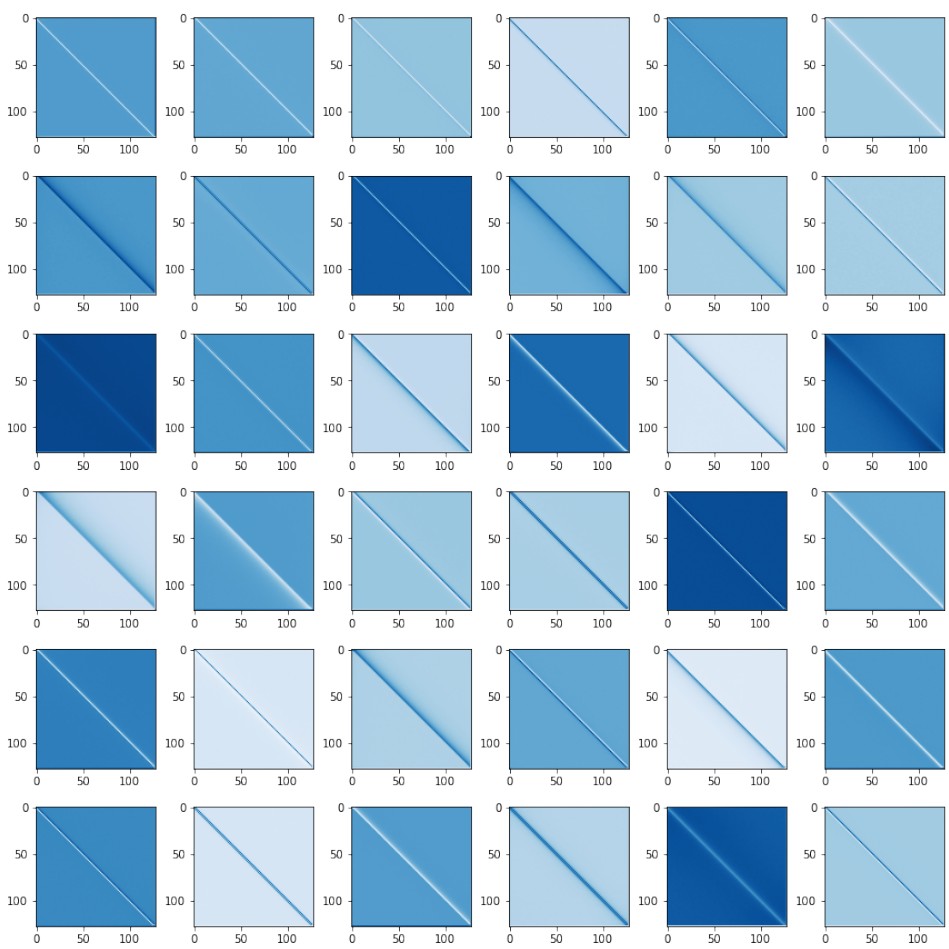

Figure 9: Spatial projection matrices learned on the MLM pretraining task without the shift invariance prior (that each individual $W$ being a Toeplitz matrix). The plots show that gMLP learns Toeplitz-like matrices (hence the notion of shift invariance) regardless.

Creating a Toeplitz Matrix (used in MLM experiments)

```
def create_toeplitz_matrix(n):
  w = tf.get_variable(
    "weight",
    shape=[2 * n - 1],
    initializer=WEIGHT_INITIALIZER)
  r = w.shape[0].value // 2
  t = tf.pad(w, [[0, n]])
  t = tf.tile(t, [n])
  t = t[:-n]
  t = tf.reshape(t, [n, n + w.shape[0] - 1])
  return t[:, r:-r]
```

## D  Visualizing Tiny Attention

Here we visualize the attention maps of the tiny attention modules in aMLP, after finetuning on MNLI-m. Each element in the heatmap below denotes the maximum attention weight of the corresponding token pair ever received during the first half of the network.

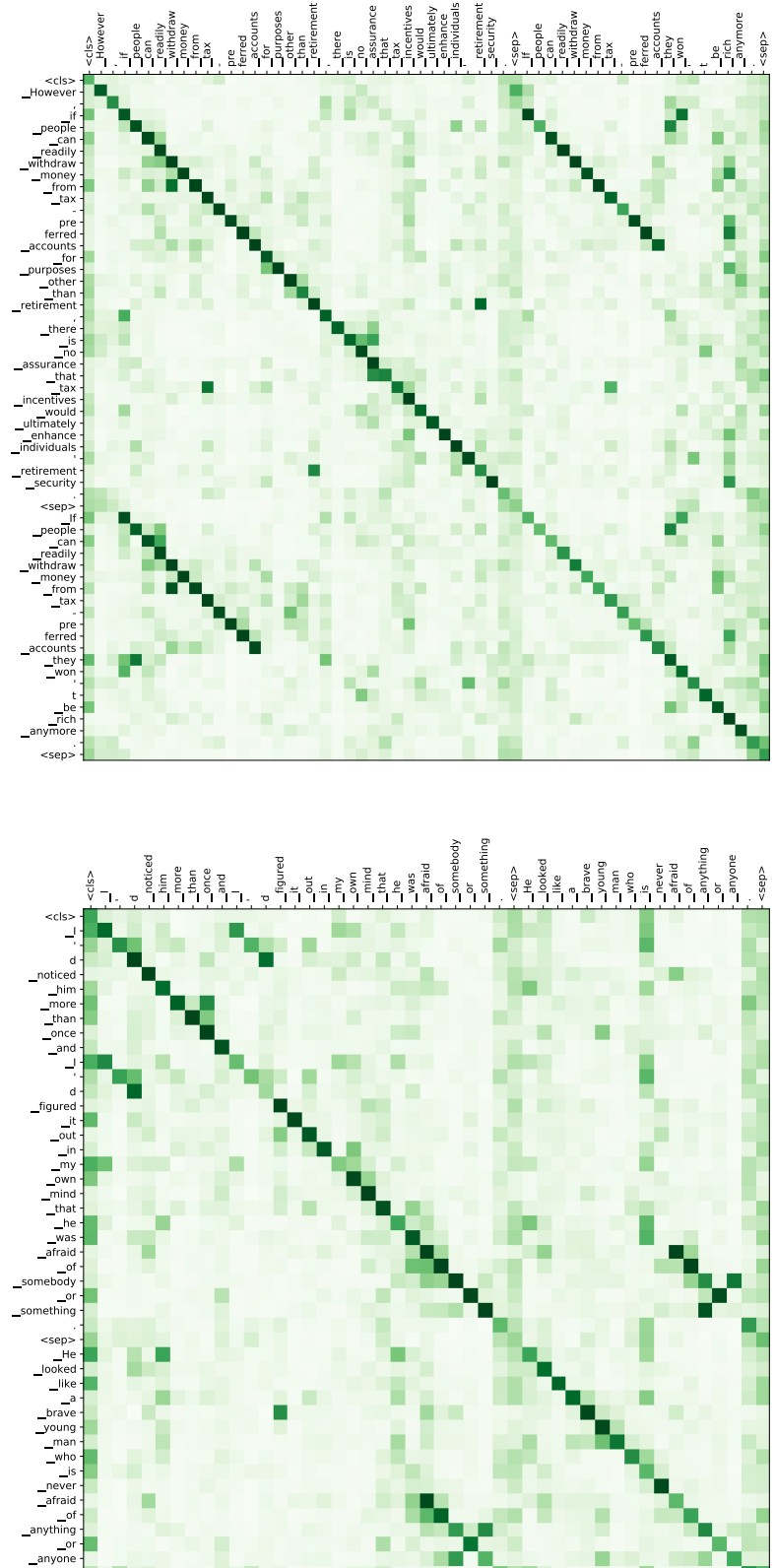

Figure 10: Attention maps in aMLP over selected examples in MNLI-m.

# E  Additional Vision Experiments

Below we present additional experiments for gMLPs on vision tasks. Specifically, we investigate a standard pretraining-finetuning setting for image classification (from $224^2$@ImageNet-21K to $384^2$@ImageNet-1K), as well as a simple object detection setup on COCO [46] with input size jittering enabled. The first setting aims to verify that gMLP models can transfer to a different image resolution in downstream tasks; the second aims to verify that gMLPs can perform well on vision tasks other than classification, and are able to handle varying input sizes during the course of training.

## E.1  Pretraining using ImageNet-21K

First, we pretrain a gMLP-Large model (#L=30, $d_{\text{model}}$=1024, $d_{\text{ffn}}$=6144) on ImageNet-21K over image size 224x224 using training settings comparable to ViT-L. Next, we "extrapolate" the 224x224 matrix for each spatial projection into a 384x384 one—this is achieved by tiling the learned spatial filters for the pixel located at the centroid of the 224x224 image. Finally, we finetune the spatially expanded model on ImageNet-1K with image size 384x384. Table E.1 below shows that gMLPs can generalize to this larger image resolution and achieve competitive results with Transformers.

| Model | ImageNet Top-1 (%) | Params (M) | Input Resolution |
|---|---|---|---|
| ViT-B/16 | 84.6 | 86 | $224 \mapsto 384$ |
| gMLP-B/16 | 84.5 | 81 | $224 \mapsto 384$ |
| ViT-L/16 | 85.1 | 307 | $224 \mapsto 384$ |
| gMLP-L/16 | 85.2 | 294 | $224 \mapsto 384$ |

Table 11: ImageNet-1K finetuning accuracies with ImageNet-21K pretraining.

## E.2  Object Detection on COCO

To get rid of the necessity of various regularization tricks, we investigate two tiny models with comparable sizes: DeiT-Tiny and gMLP-Tiny (~5M params), and use them to replace the ConvNet backbone in EfficientDet-D0 [47]. To avoid other confounding factors (e.g., ConvNet-like inductive biases), we use the vanilla ViT architecture layout *without* any spatial pooling or local shifted windows [10, 48, 38]. We apply bilinear upsampling to "reinterpret" the DeiT/gMLP endpoints as the inputs to the Bi-FPN [49]. Identical hyperparameters are used to train both models (AdamW optimizer, weight decay 0.05 and learning rate 1e-3). We use max input size 512 and apply large scale jittering to aggressively vary the effective image sizes during training. Specifically, each image is randomly resized by 0.1x-2.0x before padding/cropping [47]. Under this setting, DeiT-Tiny achieves 24.5 box mAP whereas gMLP-Tiny achieves 27.8 box mAP. While both are lower than the original EfficientNet-B0 backbone (34.6 mAP), it offers strong evidence that gMLPs are no worse than Transformers in object detection and in handling variable image sizes during training.