# OpenReview forum: "Pay Attention to MLPs"
_NeurIPS.cc/2021/Conference — NeurIPS 2021 Poster_

### Official Review · Reviewer_s4GZ · 2021-07-15

**Rating:** 6
**Confidence:** 4

**Summary:**

This paper proposes a simple mechanism: gated MLP and questions the necessity of self attention.
gMLP achieves comparable performance as DeiT on the classification task. Experiments on NLP talk also support the validness of this method.



**Limitations And Societal Impact:**

See weakness 1 of the main review.

**Main Review:**

Strength:
1. Experiments are designed across both vision classification and NLP tasks.
2. Overall, the experiment results are good.  It achieves similar performance as DeiT on ImageNet 1k classificaition and NLP baseline. gMLP achieves better performance than the MLP baseline MLPmixer.
3. The motivation is clear and good. However, the claim is too strong because of the weakness.

Weakness:
1. One big concern  is that the gMLP cannot scale to down stream tasks such as object detection, where different resolution is required during the training.  The spatial_gating_unit function in the pseudo code can only handle fixed length.
 I believe this is one fatal and fundamental drawback and would greatly limit the application of this paper. Note that self attention can handle it easily, and there are a lot of new work [1,2,3] before NeurIPS, which can easily handle this case and achieve impressive results on segmentation and object detection.   In contrast, the gMLP framework cannot deal with it directly.
Please correct me if I'm wrong.

2. Some important factors are neglected and maybe leads to unfair comparisons. One important baseline is DeiT, however, many improvements [1,2,3,4,5,6,7] are proposed based on DeiT and further boost the performance. The performance gap between transformer and gMLP are enlarged if they are included. For example,  gMLP utilizes global average pooling GAP in classification, while DeiT doesn't. This will boost transformer about at least 1-1.5% accuracy on ImageNet validation dataset [4].  And fine-tuning the drop path rate (gMLP also do this) [2, 3, 7] can marginally improve the performance of transformer.  I believe proper discussions and comparisons with these work would benefit this paper and community.

Minors:
1. The gated title is a bit confusing since it doesn't behave like  commonly used meaning.
2. Inference speed (imgs/s) is not reported (Table 2)



Reference

[1] Pyramid Vision Transformer: A Versatile Backbone for Dense Prediction without Convolutions,
 https://arxiv.org/abs/2102.12122

[2]Twins: Revisiting the Design of Spatial Attention in Vision Transformers, https://arxiv.org/abs/2104.13840

[3] Swin Transformer: Hierarchical Vision Transformer using Shifted Windows https://arxiv.org/abs/2103.14030

[4] Conditional Positional Encodings for Vision Transformers, https://arxiv.org/abs/2102.10882

[5]Tokens-to-Token ViT: Training Vision Transformers from Scratch on ImageNet https://arxiv.org/abs/2101.11986

[6] Transformer in Transformer, https://arxiv.org/abs/2103.00112.

[7] Going deeper with Image Transformers https://arxiv.org/abs/2103.17239



**Time Spent Reviewing:**

7

---

> ### Author Response · Authors · 2021-08-10
> **Response to Reviewer s4GZ**
>
> Thank you for the reviews and useful comments.
>
> -------------------
> > One big concern is that the gMLP cannot scale to down stream tasks such as object detection, where different resolution is required during the training ... I believe this is one fatal and fundamental drawback and would greatly limit the application of this paper.
>
> First, gMLPs can handle variable lengths for text. Note that all of our NLP results on GLUE (SST-2 and MNLI) and SQuAD are obtained over text sequences with variable lengths. This is achieved by applying paddings to the inputs, a common practice for Transformers in the BERT setup. To us this is a strong indication that gMLPs are applicable to a variety of high-value NLP applications including language modeling, question answering and machine translation.
>
> Moreover, gMLPs can handle variable image resolutions for image classification. Here we provide additional results as a proof of concept. Specifically, we (1) pretrain gMLP on ImageNet-21K with image size 224x224, (2) for each spatial projection, "extrapolate" the 224x224 matrix into a 384x384 matrix (this is achieved by tiling the learned spatial kernels), and then (3) fine-tune the expanded model on ImageNet-1K with image size 384x384. Table below show that gMLPs can well-handle this resolution change, achieving competitive results with Transformers:
>
> |   Model   | Params (M) | Resolution | ImageNet Top-1 (%) |
> |:---------:|:----------:|:----------:|:------------------:|
> |  ViT-B/16 |     86     |  224->384  |        84.6        |
> | gMLP-B/16 |     81     |  224->384  |        84.5        |
> |  ViT-L/16 |     307    |  224->384  |        85.1        |
> | gMLP-L/16 |     294    |  224->384  |        85.2        |
>
> Finally, we agree that handling variable-sized inputs is less straightforward for modern MLP-like models than Transformers in general. Note the former are still in their early stage. However, given the encouraging signals above in both NLP and vision, and given the success of other recent MLP-like models in segmentation and object detection (e.g., https://arxiv.org/abs/2107.08391), we believe this is rather an interesting future direction with many exciting research opportunities.
>
> -------------------
> > gMLP utilizes global average pooling GAP in classification, while DeiT doesn't.
>
> This is not true. We do not use global average pooling at all and our input/output protocols are identical with DeiT. Please refer to our model description in L70-75.
>
> -------------------
> >The performance gap between transformer and gMLP are enlarged if they (additional augmentations) are included
>
> We emphasize that none of the augmentations we applied (Appendix A.1) is beyond what DeiT have investigated (Table 8 in https://arxiv.org/pdf/2012.12877.pdf).
> To further confirm that our hyperparameter configuration did not give us any unfair advantage, we trained DeiT-B in our codebase (with droppath rate re-tuned) and obtained 81.9% on ImageNet. The difference over 81.8% (reported by the DeiT paper) is essentially within variance. We’ll include this additional result in the revised version of our paper.

---

> > ### Comment · Reviewer_s4GZ · 2021-08-16
> > **Performance of gMLP on COCO detection task, where the input resolution is changing during the training.**
> >
> > Thanks for the reply.
> >
> > I do understand the 224->384 setting on the ImageNet 1k classification task. It's still a fixed setting, and I believe extrapolating the 224x224 matrix into a 384x384 matrix works. However, the concern I mentioned in the initial review is  down stream tasks such as object detection, where different resolutions (for example, randomly sampled from 480 to 800, https://github.com/SwinTransformer/Swin-Transformer-Object-Detection/blob/master/configs/swin/mask_rcnn_swin_tiny_patch4_window7_mstrain_480-800_adamw_1x_coco.py) are required (not fixed) during the training.  I still have concerns about whether gMLP can handle it.
> >
> > The code of swin transformer was released before the DDL of the submission. It's interesting to see the performance of gMLP on object detection aligned with the coco training setting.

---

> > > ### Author Response · Authors · 2021-08-26
> > > **Object detection results with large scale jittering**
> > >
> > > Thank you for the clarification and useful suggestion. It is challenging for us to leverage the Swin Transformer implementation due to infrastructure constraints. We therefore conducted object detection experiments on COCO using the EfficientDet codebase.
> > >
> > > To get rid of the necessity of various regularization tricks, we investigate two tiny models with comparable sizes: DeiT-Tiny and gMLP-Tiny (~5M params), and use them to replace the ConvNet backbone in EfficientDet-D0. To avoid other confounding factors (e.g., convnet-like inductive biases), we use the vanilla ViT architecture layout without any spatial pooling or local shifted windows (https://arxiv.org/abs/2103.14030). We apply bilinear upsampling to "reinterpret" the DeiT/gMLP endpoints as the inputs to the Bi-FPN (https://arxiv.org/abs/2012.09958). Identical hyperparameters are used to train both models (AdamW optimizer, weight decay 0.05 and learning rate 1e-3). We use max input size 512 and apply large scale jittering to aggressively vary the effective image sizes during training. Specifically, each image is randomly resized by 0.1x-2.0x before padding/cropping (https://arxiv.org/abs/1911.09070).
> > >
> > > Under this setting, DeiT-Tiny achieves 24.5 box mAP whereas gMLP-Tiny achieves 27.8 box mAP. While both are lower than the original EfficientNet-B0 backbone (34.6 mAP), it offers strong evidence that gMLPs are no worse than Transformers in handling variable input sizes for object detection tasks during training. The new detection results also reinforce our main findings in NLP and image classification that self-attention is not key to the success of Transformers. As a next step, we plan to look into object detection under larger-scale settings and will try to include more comprehensive results on this topic in the revised manuscript.
> > >
> > > Please let us know if your concerns have been addressed.

---

> > > > ### Comment · Reviewer_s4GZ · 2021-08-27
> > > > **Not very fair comparison. Eager to see an extra experiment.**
> > > >
> > > > Thanks for the reply and experiment.
> > > >
> > > > The experiment is not quite fair. DeiT can handle changing input sequence because the underlying self attention can automatically adapt. However, its performance on downstream task is greatly weakened by improper positional encoding, i.e. the vanilla positional encoding (bicubic or bilinear interpolation) [1,2,3].   Therefore, the vanilla DeiT without proper positional encoding is a too weak baseline. And the winning of  gMLP-Tiny is not  that fair.
> > > >
> > > > Can you compare  DeiT-tiny + proper positional encoding (PEG in [2]) and PVT-tiny ([1,3]) with gMLP-Tiny on COCO using 1x schedule or 3x schedule? The PVT tiny  result (v1 and v2) on COCO is known.
> > > >
> > > > Reference
> > > > [1]Pyramid Vision Transformer: A Versatile Backbone for Dense Prediction without Convolutions, https://arxiv.org/abs/2102.12122, (ICCV21)
> > > > [2] Conditional Positional Encodings for Vision Transformers, https://arxiv.org/abs/2102.10882
> > > > [3] PVTv2: Improved Baselines with Pyramid Vision Transformer https://arxiv.org/abs/2106.13797

---

> > > > > ### Author Response · Authors · 2021-08-31
> > > > > **Confused but more results nevertheless**
> > > > >
> > > > > We're confused about the new experiment requests and wonder if there was a misunderstanding. The reviewer's original question was on handling variable image sizes in detection. In our previous response, we showed that gMLPs (which do not use position encodings) can work well in such scenarios relative to standard ViT/DeiT with vanilla position encodings (same architecture as in the ViT/DeiT papers). It is unclear to us why enhancing the baseline with a more advanced and separately developed position encoding scheme as suggested could improve the fairness of this study.
> > > > >
> > > > > Nevertheless, we conducted additional experiments with PEG layers in our EfficientDet setup, with both max image size 512^2 and 1024^2. As before, large scale jittering is enabled to vary the effective image sizes during training:
> > > > >
> > > > > |                 Backbone                | COCO mAP |
> > > > > |:---------------------------------------:|:--------:|
> > > > > |                DeiT-Tiny                |   24.5   |
> > > > > |                gMLP-Tiny                |   27.8   |
> > > > > |             DeiT-Tiny + PEG             |   28.1   |
> > > > > |             gMLP-Tiny + PEG             |   29.5   |
> > > > > | DeiT-Tiny + PEG (max image size 1024^2) |   31.6   |
> > > > > | gMLP-Tiny + PEG (max image size 1024^2) |   34.4   |
> > > > >
> > > > > Consistent with our findings earlier, the new results confirm that gMLPs can handle variable image sizes in detection without issues either with or without position encodings.
> > > > >
> > > > > Finally, we'd like to emphasize that our main contribution is to reveal that self-attention is not critical for the success of Transformers. We believe this claim has been backed up by extensive experiments in NLP (with variable text lengths) and image classification already, which are the primary tasks investigated in the seminal papers of BERT and ViT. While we tried our best to provide additional results on object detection during the rebuttal, we'd like to leave further explorations on this specific task and its implications on position encodings as future work.

---

> > > > > > ### Comment · Reviewer_s4GZ · 2021-08-31
> > > > > > **Post Feedback about the extra experiment.**
> > > > > >
> > > > > > Thanks for the extra experiment.
> > > > > > My concern is damped, therefore, I raise my score to Marginally above the acceptance threshold.
> > > > > >
> > > > > > I hope you discuss the performance of gMLP on detection in the revised version. After all, transformer based backbones [1,2,3] all outperform convolutional networks with clear margins on detections.  The efficientdet with gMLP experiment seems a bit counterintuitive.   It would be better to discuss it. A good work should point out its limitations as well.
> > > > > >
> > > > > > Reference
> > > > > >
> > > > > > [1] Pyramid Vision Transformer: A Versatile Backbone for Dense Prediction without Convolutions, https://arxiv.org/abs/2102.12122
> > > > > >
> > > > > > [2]Twins: Revisiting the Design of Spatial Attention in Vision Transformers, https://arxiv.org/abs/2104.13840
> > > > > >
> > > > > > [3] Swin Transformer: Hierarchical Vision Transformer using Shifted Windows https://arxiv.org/abs/2103.14030

---

### Official Review · Reviewer_Upxb · 2021-07-16

**Rating:** 8
**Confidence:** 4

**Summary:**

This paper proposes and studies an interesting research problem whether we can remove the self-attention in Transformer based models. The authors devise a novel architecture called gMLPs by removing the self-attention layer and introducing a new Spatial Gating Unit into the FFN module to well model the interactions between independent tokens. Comprehensive experiments on CV or NLP tasks confirmed that the proposed gMLPs can achieve comparable performances with the standard ViT or BERT models.

**Limitations And Societal Impact:**

I see no potential negative societal impact of this work.

**Main Review:**

I enjoyed reading this paper, the whole paper is very clear and well-written, and I believe this paper is going to be useful for the community and inspire other researchers’ work on this topic.

Below are some questions or confusions I had:

1. The proposed architecture are evaluated on encoder-style pre-trained models, e.g. ViT or BERT, will it be possible to extend the method to encoder-decoder or decoder-style models, like T5 or GPT?

2. Transformer-based models tend to have better OOD generalization ability, the authors are suggested to add some OOD evaluation experiments as conducted in the paper “Pretrained Transformers Improve Out-of-Distribution Robustness”.

3. In Tables 4 and 5, it is better to add inference time and flops as extra evaluation metrics.

4. In Figure 4, it seems that the learned filters tend to have local correlations, will this limit the ability of gMLPs in modeling long range dependencies.

5. In appendix D the learned spatial projection matrices seems to be Toeplitz-like, can you quantitatively evaluate how these matrices be Toeplitz-like?

6. Will the proposed gMLPs model be sensitive to the length of input sentence? The authors are suggested to add some analysis with considering the sequence length factor, e.g. accuracy vs length on downstream tasks.

7. In practice, before deploying to products, pre-trained models often need to be compressed, I wonder whether the gMLPs can be compressed into some tiny models for IoT devices.

8. It is better to discuss some potential limitations of the proposed gMLPs.


**Time Spent Reviewing:**

three hours

---

> ### Author Response · Authors · 2021-08-10
> **Response to Reviewer Upxb**
>
> Thank you for the reviews and valuable suggestions.
>
> --------------
> > will it be possible to extend the method to encoder-decoder or decoder-style models, like T5 or GPT?
>
> Yes. We conducted additional language modeling experiments on C4, and our preliminary results indicate gMLPs are able to achieve comparable perplexities with Transformers in the decoder-only setting too (using a GPT-2-like architecture layout). The only change we made relative to our MLM model is to mask out the lower-triangular part of the spatial projection matrix in gMLPs to prevent causal information leak. We will add more discussions for decoder-style applications in the revised paper.
>
> --------------
> > the authors are suggested to add some OOD evaluation experiments as conducted in the paper "Pretrained Transformers Improve Out-of-Distribution Robustness".
>
> We agree a robustness analysis regarding OOD data would be very interesting (since a different model family is proposed) and we will look into the work suggested.
>
> --------------
> > In Tables 4 and 5, it is better to add inference time and flops as extra evaluation metrics.
>
> Acknowledged. We'll add them in the revised paper.
>
> --------------
> > In Figure 4, it seems that the learned filters tend to have local correlations, will this limit the ability of gMLPs in modeling long range dependencies.
>
> This is possible and may explain why gMLPs perform well especially when they are deep enough (Table 4). On the other hand, it is also interesting to notice that the learned receptive fields in Figure 4 are in general wider than what people typically would use in ConvNets (i.e., 3, 5, 7).
>
> --------------
> > In appendix D the learned spatial projection matrices seems to be Toeplitz-like, can you quantitatively evaluate how these matrices be Toeplitz-like?
>
> One way to quantify this is to measure the absolute error rate of the learned position-specific filters relative to a shared position-agnostic filter (obtained by taking the mean of position-specific filters). Zero error is expected if the learned filters are perfectly spatial invariant (corresponding to a Toeplitz spatial matrix). We take the last layer of gMLP-base and report its filter error rates with increasing window size d: 1.4% (d=5), 2.4% (d=9), 7.0% (d=17), 14% (d=33) and 18% (d=65). The results indicate that the error rates are in general small, and that the elements near the diagonal of the matrix (which usually carry more weights) can be more accurately approximated by a Toepltz matrix.
>
> --------------
> > Will the proposed gMLPs model be sensitive to the length of input sentence?
>
> One related experiment we tried on GLUE is to feed gMLP models with longer, redundant input sequences (constructed by repeating the original input sequence multiple times). We also tried inserting a large number of <sep> tokens in the middle of the premise segment and the hypothesis segment on MNLI (to obtain a longer sequence with the same semantic meaning). Interestingly, neither of these leads to any significant change in the output behavior of gMLPs.
>
> --------------
> > I wonder whether the gMLPs can be compressed into some tiny models for IoT devices.
>
> One interesting direction along this line is to train a large Transformer/gMLP teacher and then distill it into a tiny gMLP student for deployment. We believe the simplicity of gMLP could make it desirable for edge chips specialized in basic matmuls instead of the more generic einsums (required for handling the extra head dimensions in Transformers).
>
> --------------
> > It is better to discuss some potential limitations of the proposed gMLPs.
>
> Good point -- we'll add more discussions in the revised version. One noticeable aspect that we've discussed is that gMLPs seem to carry a different type of inductive bias from Transformers, making them better on certain NLP downstream tasks but worse on some others (Figure 5). This can be addressed by incorporating a small single-head self-attention (Figure 7), which we believe is primarily responsible for cross-segment alignment (Appendix E).

---

> > ### Comment · Reviewer_Upxb · 2021-08-31
> > **About inference time**
> >
> > Thanks for the reply.  The potential application scenarios highly depend on the inference time and storage consumption of the proposed model, so can you provide some preliminary results on the inference time for different tasks with various input lengths?

---

> > > ### Author Response · Authors · 2021-09-02
> > > **Inference costs wrt input lengths**
> > >
> > > Thank you for the question. Below we report inference latencies on V100 GPUs wrt various input sizes.
> > >
> > > *Text classification (gMLP-base)*:
> > >
> > > |  Seq length  | 32 | 64 | 128 | 256 | 512 | 1024 | 2048 | 4096 |
> > > |:-----------------:|:--:|----|-----|-----|-----|------|------|------|
> > > | V100 Latency (ms) | 13 | 13 | 14  | 22  | 31  | 65   | 148  | 386  |
> > >
> > > *Image classification (gMLP-B/16)*:
> > >
> > > |     Resolution    | 56x56 | 112x112 | 224x224 | 448x448 | 986x896 |
> > > |:-----------------:|:-----:|:-------:|:-------:|:-------:|:-------:|
> > > | V100 Latency (ms) |   7   |    7    |    12   |    30   |   158   |
> > >
> > > While the inference latency can be highly hardware-dependent, it is still interesting to empirically observe its relationship with the input size in this case: the latency grows sublinearly at the beginning and then superlinearly when the input size becomes large. This is because the model's cost mostly comes from the channel expansion/projection ops (O(nd^2)) initially, but then gradually becomes dominated by the spatial projection (O(dn^2)) as the input length n increases. The results above are very preliminary due to the limited time left for the discussion period, but we will try to include a more comprehensive analysis in the revised manuscript.

---

### Official Review · Reviewer_5t8k · 2021-07-19

**Rating:** 6
**Confidence:** 5

**Summary:**

This paper presents a simple architecture based on MLPs, which only adopts channel projections and spatial projections with the spatial gating units. On image classification, the presented gMLP could achieve on par performance with DeiT and ViT, and better than previous MLP-based approaches, such as MLP Mixer. On masked language modeling tasks, gMLP could also achieve on par performance with Transformer on Perplexity, and gMLP could perform slightly worse than Transformer on MNLI.

**Limitations And Societal Impact:**

Yes.

**Main Review:**

For originality, this paper presents a new MLP-based approach, e.g. gMLP. But compared to MLP-Mixer, the contribution of this gMLP seems marginal. Only the spatial gating unit is newly designed and verified to be effective. But the experimental study of this paper is sufficient and valuable to the community.

The quality of this paper is good. From an experimental report perspective, this paper contains sufficient experimental results, verifies important claims, and achieves relatively good performance on both ImageNet classification and masked language modeling in NLP, comparing to the previous approaches. The diversity of the experiments is sufficient.

The clarity of this paper is good. This paper is well-organized and easy to follow.

My major concern is about spatial projection. This is a simple design, but the application scenario is also limited due to the fixed size of the input spatial resolution. For example, for some vision downstream tasks of object detection and semantic segmentation, the input sizes are different from that of the image classification. This architecture could not be directly applied to these tasks.

**Time Spent Reviewing:**

10

---

> ### Author Response · Authors · 2021-08-10
> **Response to Reviewer 5t8k**
>
> Thank you for the reviews and useful feedback.
>
> -------------------
> > But compared to MLP-Mixer, the contribution of this gMLP seems marginal.
>
> We respectfully disagree and would like to point out that MLP-Mixer, FF and ResMLP (released on arXiv near the NeurIPS deadline) are **concurrent works with ours**. We cited them and included their results as an effort to provide the readers a comprehensive overview of those very recent MLP-like models. This, however, does not mean that our submission (which was independently developed) should be evaluated as a subsequent work.
>
> We also make an important contribution in NLP which MLP-Mixer did not. In particular, we found that MLP-Mixer does not perform well on language tasks (Table 3), potentially due to the lack of higher order interactions across tokens. By leveraging the concept of spatial gating, gMLP is the first MLP-like model which can compete with Transformers on masked language modeling. This alone is an important contribution considering the impact of BERT in NLP.
>
> -------------------
> > My major concern is about spatial projection. This is a simple design, but the application scenario is also limited due to the fixed size of the input spatial resolution.
>
> First, gMLPs can handle variable lengths for text. Note that all of our NLP results on GLUE (SST-2 and MNLI) and SQuAD are obtained over text sequences with variable lengths. This is achieved by applying paddings to the inputs, a common practice for Transformers in the BERT setup. To us this is a strong indication that gMLPs are applicable to a variety of high-value NLP applications including language modeling, question answering and machine translation.
>
> Moreover, gMLPs can handle variable image resolutions for image classification. Here we provide additional results as a proof of concept. Specifically, we (1) pretrain gMLP on ImageNet-21K with image size 224x224, (2) for each spatial projection, "extrapolate" the 224x224 matrix into a 384x384 matrix (this is achieved by tiling the learned spatial kernels), and then (3) fine-tune the expanded model on ImageNet-1K with image size 384x384. Table below show that gMLPs can well-handle this resolution change, achieving competitive results with Transformers:
>
> |   Model   | Params (M) | Resolution | ImageNet Top-1 (%) |
> |:---------:|:----------:|:----------:|:------------------:|
> |  ViT-B/16 |     86     |  224->384  |        84.6        |
> | gMLP-B/16 |     81     |  224->384  |        84.5        |
> |  ViT-L/16 |     307    |  224->384  |        85.1        |
> | gMLP-L/16 |     294    |  224->384  |        85.2        |
>
> Finally, we agree that handling variable-sized inputs is less straightforward for modern MLP-like models than Transformers in general. Note the former are still in their early stage. However, given the encouraging signals above in both NLP and vision, and given the success of other recent MLP-like models in segmentation and object detection (e.g., https://arxiv.org/abs/2107.08391), we believe this is rather an interesting future direction with many exciting research opportunities.

---

### Official Review · Reviewer_GiR7 · 2021-07-21

**Rating:** 6
**Confidence:** 4

**Summary:**

This paper proposes a MLP-based model that achieves comparable performance with transformers for vision tasks and language modeling. While this MLP-based model performs slighly worse on natural language understanding tasks (e.g. MNLI), introducing a small single-head self-attention module can compenate for most of the performance loss. Compared to other MLP-based models introduced in concurrent works, the proposed model uses a gating module before the spatial projection. Performance wise, the proposed method is stronger than the concurrent MLP variant MLP-mixer on vision and language modeling tasks, showing the advantage of the proposed gating methods

Overall I find the proposed method to be interesting and empirically strong. I am leaning toward accept but also wish the authors had done more work in understanding the proposed models. I offer some suggestions and have some questions below.

**Ethical Concerns:**

I don't have any ethical concern with this work.

**Ethics Review Area:**

["I don’t know"]

**Limitations And Societal Impact:**

I don't see any negative societal impact from this work.

**Main Review:**

Pros

- The empirical performance is strong
- The proposed model is novel in the sense that it combines a gating mechanism with MLP-only model.
- The submission is well-written and clear.

Cons/Questions

- The submission very much focuses on the empirical performance of the proposed method. This is understandable but I also hope the authors had done more analysis work to understand the proposed methods. Here are some suggestions:
    - Learning dynamics: Are the proposed method noticeably different from transformers in terms of convergence behavior?
    - Data efficiency: Is there trade-off between transformers and gMLP in terms of data efficiency? For example, does gMLP perform better or worse relative to transformer as we add in more training data?
- Can the authors document the computational advantage of gMLP? Is the proposed method faster to inference or does gMLP save GPU memory?
- The fact that attention mechanism is especially important for cross-sentence understanding tasks is very interesting. I wonder if the authors can identify any patterns in the learned single head attention. Or, can the authors offer more discussion on how the attention mechanism is helping?

**Time Spent Reviewing:**

6

---

> ### Author Response · Authors · 2021-08-10
> **Response to Reviewer GiR7**
>
> Thank you for the reviews and questions.
>
> ----------------------
> > Learning dynamics: Are the proposed method noticeably different from transformers in terms of convergence behavior?
>
> They converge equally fast for masked language pretraining. As for image classification, gMLP empirically converges faster than DeiT (ViT + regularization) under comparable training settings on ImageNet. Below we report their validation accuracies across steps:
>
> |                     | step-30K | step-60K | step-92K (end of training) |
> |---------------------|-----------|:---------:|:---------------------------:|
> | DeiT-B (87M params) | 74.7      |    80.2   |             81.9            |
> | gMLP-B (73M params) | 78.3      |    81.2   |             81.8            |
>
> This is aligned with our main finding that multi-head self-attention is not a necessary component for efficient learning. We’ll include more discussions in the revised paper.
>
> ----------------------
> > Data efficiency: Is there trade-off between transformers and gMLP in terms of data efficiency?
>
> Good question. In Figure 5 we’ve shown that the two models scale equally well over increased parameters. Below we further show that they are on par in scalability over increased data.
>
> Vision results: enlarging the training set from ImageNet-1K to ImageNet-21K:
>
> | Model        | Training Data | Inference Params (M) | ImageNet-1K Top-1 (%) |
> |--------------|---------------|:--------------------:|:---------------------:|
> | ViT-B (+reg) | ImageNet-1K   |          86          |          81.8         |
> |    gMLP-B    | ImageNet-1K   |          73          |          81.6         |
> |     ViT-B    | ImageNet-21K  |          86          |          84.6         |
> |    gMLP-B    | ImageNet-21K  |          81          |          84.5         |
>
> NLP results: enlarging the amount of training tokens by 16x on C4:
>
> |        | Training tokens     | Params (M) | SST-2 | MNLI-m |
> |--------|---------------------|:----------:|:-----:|:------:|
> | BERT-L | 2K bsz x 125K steps |     336    |  94.3 |  87.0  |
> | gMLP-L | 2K bsz x 125K steps |     365    |  94.8 |  86.2  |
> | BERT-L | 8K bsz x 500K steps |     336    |  95.3 |  88.9  |
> | gMLP-L | 8K bsz x 500K steps |     365    |  96.1 |  88.0  |
>
> ----------------------
> > Can the authors document the computational advantage of gMLP?
>
> There are several potential computational advantages of gMLPs which we didn't leverage in our implementation. For instance, each spatial projection in gMLPs can in theory be performed in O(n log(n)) MAdds via Fast Fourier Transform (FFT) when the matrix is Toeplitz (which empirically appears to be the case: L142-149). This is better than Transformer’s multi-head self-attention with O(n^2) MAdds especially for long input sequences. The simplicity of gMLPs also makes them desirable for model parallelism, whereas Transformer’s multi-head self-attention modules can be tricky to get partitioned efficiently across workers.
>
> ----------------------
> > I wonder if the authors can identify any patterns in the learned single head attention. Or, can the authors offer more discussion on how the attention mechanism is helping?
>
> Please refer to Appendix E for visualizations of the single-head attention. There‘re strong indications that the tiny attention is doing superficial alignment between the input sentence pairs--note the clear stripes on the off-diagonal of the attention matrices.

---

### Decision · Program_Chairs · 2021-09-27

**Decision:**

Accept (Poster)

**Comment:**

The paper introduces an MLP-based model with gating (gMLP) that achieves comparable performance to Transformers, showing that self-attention is not critical for the success of these models. While multiple concerns were raised regarding the limitations of the proposed method, all four reviewers appreciate the strong empirical results reported in the paper and recommend acceptance. The AC agrees with this decision, and requests the authors to add to the final version the discussion and additional information provided in the rebuttal, as well as more clearly describe the limitations of the proposed approach.